# Psychoanalytic Interventions with Abusive Parents: An Opportunity for Children’s Mental Health

**DOI:** 10.3390/ijerph192013015

**Published:** 2022-10-11

**Authors:** Anna Maria Rosso

**Affiliations:** Department of Education, University of Genoa, 16128 Genoa, Italy; rosso@unige.it

**Keywords:** physical child abuse, psychoanalytic intervention, transgenerational transmission of abuse, child–parent psychotherapy

## Abstract

Research has extensively shown that most people who experience maltreatment in their childhood develop mental disorders, psychosocial adjustment problems, and, in many cases, become maltreating adults themselves. Preventing child maltreatment and treating abused children and abusive parents are, therefore, pressing public health issues. As established by the UK Children Act in 1989, child development is enhanced by remaining in the family whenever the child’s safety is assured. Thus, developing prevention and intervention programs for the purpose of repairing, whenever possible, the child–parent relationship should be a social priority. This narrative review focuses on the psychoanalytic studies related to intrapsychic dynamics and therapeutic intervention for physically abusive parents. The role of the transgenerational transmission of abuse and parents’ narcissistic fragility is crucial. Psychoanalytic interventions focus on helping the parent work through their past painful experiences and narcissistic vulnerability. Parent–child psychotherapy and mentalization-based treatment have been found to be prevalent, while there is scarce literature regarding intensive individual psychoanalytic treatment. Within the framework of attachment theory, brief interventions were developed; however, they did not prove effective for those parents who suffered experiences of maltreatment or severe neglect in childhood and for whom long-term parent–child psychotherapy resulted, which proved to be the most effective.

## 1. Introduction

A considerable body of research has profusely demonstrated that the long-term effects of child maltreatment are devastating: many people who have been abused in their childhood develop later mental disorders, manifest psychosocial adjustment problems, and, in many cases, become abusive adults themselves [1,2]. Preventing child maltreatment and treating abused children and abusive parents are, therefore, pressing public health issues. 

When a child is at risk of abuse in a family or when abuse has already occurred, assessing whether the child should be removed from the family or whether intervention in the family can prevent future abuse is of the utmost importance. Even in cases where the child is initially removed from the family for his or her own protection, it is necessary to assess whether an intervention with the parents will suffice to allow the child to return to the family later. It is, therefore, a matter of assessing how motivated the parents are to engage in a path of change and which intervention is most appropriate in each specific situation.

In my experience, when intervention programs are not available for abusive parents or those parents at risk of abuse, it unfortunately often happens that the child is removed from the family to be placed in residential facilities or foster families, without careful consideration being given to the potential long-term harmful effects following the disruption of the relationship with his or her family, although these effects are extensively highlighted in the literature [3,4,5]. Indeed, while removal from the family protects the child from the risk of maltreatment, it also exposes him/her to permanently undermining the possibility of repairing the relationship with his/her parents, understood both as real persons and as his/her internal representations. As psychoanalysis has taught us extensively, living with bad internal objects implies constant struggle and suffering along with a greatly increased risk of developing psychopathological disorders and maladjustment in interpersonal relationships. Moreover, as Novick and Novick [6] stated, parents, when the relationship with them is good enough, are a lifelong resource. Whenever possible, therefore, interventions should be aimed at repairing the relationship [3,5], and social workers should avoid being tempted by the impulsive, action-oriented style of the families they work with by choosing quick and drastic solutions that may be harmful to the child in the long run [7].

As also established by the Children Act in 1989 [8], child development is enhanced by remaining in the family whenever the child’s safety is assured, and social services should strike a balance between protecting children and ensuring that they can remain with their families [9]. Developing prevention and intervention programs in public health services for the purpose of repairing, whenever possible, the child–parent relationship should be a social priority. 

Psychoanalytic studies have aimed at understanding the internal world of abusive parents and identifying what interventions might help them become more competent. Studying and comprehending the inner world of parents who are prone to physically abusing their children can be helpful in understanding the deeper motivations for their behavior as well and, thus, in devising interventions not only aimed at modifying their behavior but also directed at intervening in the unconscious dynamics that lead them to the abusive behavior. 

Since the pioneering work of Selma Fraiberg [10] at the Child Development Project, University of Michigan, in the 1970s, other psychoanalysts have worked to develop intervention models based on a psychoanalytic understanding of the mental functioning of parents who abuse their children or are at risk of doing so. It was the groundbreaking work of psychoanalysts such as Richard Galdston [11], Brandt Steele [12], and Selma Fraiberg herself [10,13] who highlighted the extent to which child abuse can be considered a treatable syndrome and the outcome of a breakdown in the parent–child relationship, which enabled the development of intervention programs aimed at repairing the relationship. 

The goal of this article is to present a narrative review of psychoanalytic studies in this field, specifically focusing on the intrapsychic dynamics of parents who physically abuse their children, and the therapeutic interventions developed by psychoanalysts to treat maltreating parents.

The article focuses specifically on physical abuse, following the recommendation [14] not to group together different types of abuse, since it has long been known that, for example, sexual abuse is an issue completely apart from physical maltreatment [3]. As it is only directed at studies that have addressed intrapsychic dynamics and the treatment of physically abusive parents from an exclusively psychoanalytic perspective, the review does not examine other types of treatments, even though they have proven effective. 

## 2. Methods

For the purposes of conducting a narrative review to present an overview of studies published to date, highlight emerging themes, and identify future research directions, the terms “child abuse or neglect or maltreatment or mistreatment” and “psychoanalysis or psychoanalytic or psychodynamic” or “infant–parent psychotherapy” and “parent*”were searched in titles, abstracts, and main texts using the Psycarticles, Psychinfo, Psychology and Behavioral Sciences Collection, Medline, and Pep-web databases. Inclusion criteria comprised original articles published in the English language in peer-reviewed journals before June 2022. According to the exclusion criteria, dissertations and chapters in books were not taken into account. After applying the exclusion criteria, 386 potentially relevant articles were found, and, finally, after examining the abstracts and bibliographies of each article, 23 papers were selected for the present review. Rather surprisingly, only about 5% of the reviewed abstracts showed that the studies focused on parents who had abused their children. Articles were excluded from the review for the following reasons: 41 addressed the topic of parents committing sexual abuse; 10 referred to the historical and/or sociological aspects of parenting; 23 were about other types of maltreatment (neglect, emotional abuse, Munchausen syndrome by proxy); 2 were about mentally ill mothers who had killed their children; 1 was about parental noncompliance in pediatric treatment; in 2 cases, the articles could not be found; and in the remaining cases, the articles either mentioned only psychoanalytic or psychodynamic interventions without addressing it or were about the psychoanalytic treatment of people who had been abused by their parents. 

## 3. Results

### 3.1. The Intrapsychic Dynamics of Parents Who Physically Abuse Their Children

#### 3.1.1. The Transgenerational Transmission of Abuse

Rather surprisingly, it was not until the 1960s that child abuse became the focus of attention in social policy and psychological studies [15]. Steele [12] highlighted the extent to which the abusive parent, closely identified with a harsh and rejecting mother and a negative self-image dating back to childhood, subjected their child to similar experiences suffered in childhood. In his clinical experience, he observed that these parents often predominantly resorted to the defense mechanisms of denial, projection, identification with the abuser, and role reversal. Green [15] conducted the first empirical study of a group of 60 abusive mothers using a comparison group of 30 neglectful mothers and a control group of 30 nonabusive parents. He found corroboration for Steele’s clinical experience and emphasized how crucial it was for abusive mothers to resort to role reversal in the expectation that their children should fulfill unmet needs for dependence and the confirmation of their goodness. The child is most likely to be abused when he/she is not gratifying to the mother, thus, when he/she is most demanding, needy, and irritated. Probably for this reason, Green [15] surmises, abuse is more frequent in the child’s first two years of life and when the child has physical defects or atypical development. The child who cries often and does not gratify the mother with affectionate and compliant behavior is experienced by abusive mothers as confirmation of their inadequacy and evokes in them the rejection perceived by their parents during childhood. 

Recent studies [2,16] have confirmed that having been abused in childhood is an important risk factor; however, studies on the transgenerational transmission of abuse have actually shown that not all abused children become abusive parents, and they have highlighted some protective factors that break the cycle of abuse: having received emotional and social support from significant adults, having received psychotherapy, and being able to acknowledge one’s feelings of anger about the abuse suffered [17]. In addition, it must be kept in mind that abuse is the result of the interaction between individual, family, social, and cultural factors [18] and that having been abused in childhood is, therefore, only one of the individual factors that may be involved. Wilkes [19] found in a group of abused parents who did not become abusers that they clearly recognized how wrong their parents’ behavior had been and attributed no blame to themselves for what had happened, and he hypothesized that these were two crucial resilience factors. Already Selma Fraiberg [13] had observed that the possibility of experiencing pain from one’s childhood history is a crucial deterrent against repeating abuse toward one’s children, and a recent study [20] showed that maintaining good memories of childhood moderates the transgenerational transmission of abuse. A recent meta-analysis [21] confirmed that sensitive, stable, and secure relationships can break the cycle of abuse. When the child does not have the opportunity to rely on other sensitive and caring adults, he/she is mostly forced to deny the malevolence of the abusive parent to protect his/her internal parental image. He/she often blames him/herself for the parent’s abusive behavior, and this seems to exert a dual function: on the one hand, seeing him/herself as bad saves the parent’s image, and on the other hand, it makes the child feel active and in control of the situation. This may lead the child to enact disturbing behaviors [22].

Moreover, as Galdston [23] states, the abusive parent–child relationship is characterized by intense ambivalence and deep attachment. The strength of this attachment makes clinical management more complicated than in cases where parents are indifferent to the child. “Parents and children who have been largely indifferent to each other may take leave with little regret. They have suffered the cold of indifference rather than the heat of anger” [23] (p. 392). Furthermore, the abused child who has taken upon him/herself the malevolence of the parent in order to maintain a good internal image of him/herself will suffer from low self-esteem, may have little autonomy, and will lack the ability to differentiate self from other [22].

Steele and Pollock [24] interviewed 60 abusive families and found that few among them had been physically abused by their parents; rather, all of the abusive parents interviewed had felt very criticized by their parents, unwanted, and unloved.

The mechanisms by which the transgenerational transmission of abuse occurs have been more clearly highlighted recently by observational studies of the parent–child relationship and empirical studies in the area of attachment theory and mentalization.

Seligman [25] reports a particularly telling observation of the interaction between a father who was repeatedly abused as a child, and his son, a three-day-old boy.

“In this brief episode, he holds his baby very awkwardly, just below his neck, and forcefully brings the neonate’s face close to his own with a look that seems to convey some tenderness along with much anxiety. Next, the father tries to force Daniel to drink from a bottle while the baby desperately shows that he is satiated, first by not sucking and keeping his mouth closed and then by tensing up and finally going limp; during this sequence, the father rebuffs efforts by his wife and a therapist-observer to get him to notice Daniel’s resistance to his brutal ministrations, remaining oblivious to his son’s repeated signals. The father again brings his face intrusively close to his son’s, calls him ‘Chump!’ and says in a pugilistic manner, ‘Do you want to tell me about it?’ He hoists the baby high up in the air, as if he were roughhousing with a much older boy. Finally, as the baby seems to collapse into a droopy, withdrawn state, the father exclaims, ‘That’s enough of your garbage!’” [25] (p. 138).

Seligman [25] observes that, no matter how characteristic this type of interaction may become in his interpersonal experience, this child, who has experienced a very distressing sense in his bodily experience of being prodded, jostled, and deprived of any comfort and control of his body, will feel that being helpless and incapacitated are fundamental modes of the experience of the self.

The observation of this interaction also provides insight into the concept of pathological projective identification. The father disregards his son’s signals, treating him with hostility while feeling that he loves him (hostility is perceived by observers). In this way, he externalizes his “bad,” powerless, hostile self by attributing it to his son, enacting this without any reflective thought. Under these conditions, Daniel can not only do nothing but identify with these emotional and relational states, including the sense of powerlessness, which the father keeps out of his awareness by pouring it onto his own son, but he can also internalize a pattern, in that the way to deal with his own feeling of powerlessness is to make the other person feel powerless.

This example also shows how it may be precisely the feeling of fragility evoked by the very young child that stimulates abusive behavior in parents who have not felt that their own fragility has been respected and have, thus, been unable to develop feelings of acceptance and tenderness for their own and others’ fragility.

Empirical studies conducted within the framework of attachment theory have fully demonstrated that the Internal Working Models (IWM) of self and other are transmitted from one generation to the next along with defensive strategies for coping with distress in the face of perceived danger. Mothers with secure attachment patterns help their children develop good emotional regulation strategies, while mothers with distancing attachment patterns emphasize autonomy and minimize distress, and mothers with preoccupied attachment patterns are unpredictable. They, alternating responsive responses with neglectful or inappropriate responses, promote emotional dysregulation, as do mothers who have developed disorganized patterns [26,27]. Interestingly, these studies highlighted the crucial role of mentalization, operationalized as reflective functioning, in the transmission of attachment security: the children of mothers who experienced negative childhood experiences but were able to come to terms with them by acknowledging their own suffering and trying to understand their behaviors and those of their parents, as motivated by mental states, more frequently developed attachment security. These studies provided important empirical support for what had been previously tested in clinical work [27].

A number of studies have specifically investigated the quality of attachment patterns and reflective function in abusive parents [28,29,30,31], highlighting the prevalence of dismissive and disorganized mind states in abusive parents. Ammaniti and colleagues [28] found that 87.5% of abusive parents evinced an insecure state of mind regarding attachment and that 47% of these insecure parents specifically exhibited unintegrated states of mind, such as Unresolved or Cannot Classify states of mind, as a consequence of dissociative states that developed following early traumatic experiences of loss and/or abuse. Rosso [30] found that, in her sample, maltreating parents had not experienced poverty, poor education, lack of social support, or physical illness in their childhood, but they had suffered high degrees of family conflicts associated with significant neglect experiences, especially with their mothers, findings already reported in previous studies [32,33].

In Rosso’s study [30], more than 90% of the abusive parents showed an insecure state of mind regarding attachment with a prevalence (66.7%) of dismissing classification. Compared with the control group of nonabusive parents, they showed high reliance on the defensive strategies of idealizing their childhood experiences, derogating their attachment needs, lacking memories in addition to passivity in thought processes, and disorganized states of mind regarding grief and/or abuse experiences. Rosso argues that the marked tendency to derogation that these parents exhibit may specifically explain how the devaluation of relationships and attachment needs—probably acquired in their childhood history to defend themselves through exclusion from awareness of emotional pain and their experiences of fragility and helplessness—makes it very difficult for them to take care of their children’s emotional needs. In this study, the dismissive and derogatory state of mind was found to be associated with severe mentalizing deficits, often with a particular type of deficit called Negative Reflective Functioning, which involves the tendency to avoid and refuse to take a reflective stance. The study found that parents who inflicted the most severe maltreatment on their children showed this specific mentalization deficit. As Fraiberg [10] already observed in her clinical work, these parents were not able to process their painful childhood experiences and were not aware of either their own emotional pain or that inflicted on their children. The results of this research stimulated the development of specific interventions centered on attachment and mentalization, as discussed below.

#### 3.1.2. Narcissistic Fragility

Many authors emphasize how parents who abuse their children suffer from significant narcissistic fragility. Eldridge and Finnican [34] pointed out that parents who abuse their children unknowingly wished to become parents for the purpose of finally having someone to maintain their narcissistic balance; in other words, to support their psychic integrity by supporting their fragile self-esteem.

Early experiences of good caregiving allow the child to feel safe and idealize caregivers, perceiving them as very powerful and eager to share and bestow their strength. This basic trust allows the child to later tolerate inevitable frustrations provided they are progressively and appropriately dosed to the child’s coping capacities. The early relationship with an empathetic parent, who is able to understand how much frustration the child can tolerate, enables the child to develop a cohesive nuclear self [35], which implies possessing an internal sense of importance and value that is stable over time and helps to comfort oneself in difficult times, as well as to set balanced ambitions and ideals.

In Eldridge and Finnican’s view, the personality of the abusive parent exhibits a particular form of developmental disorder in which the child, despite him/herself, is called upon to play the role of one who must support the narcissistically fragile parent by helping him or her maintain a sense of self. In addition, the narcissistically fragile parent experiences particularly intense difficulties when coping with parenting because his or her unmet child needs are stimulated by the task of taking care of the child’s needs. The young child demands that his own needs be met and does not gratify the parent’s needs, especially when he/she cries and when he/she is demanding. The narcissistically fragile parent, when his/her needs are frustrated, may have angry outbursts and mistreat his/her child.

Narcissistic deficits were further observed by Blumberg [3], Crivillé [36], Rosen [37], and Purcell [38]. The latter, during his decade-long work at Children’s Charter with abused children, observed that these families are often described as “upside-down”, in that they are characterized by role reversal, projection, and disturbed attachment: parents re-enact the unmet needs of their own childhood while forcing their children to assume the role of those who must support their self-esteem. When children fail to meet their parents’ needs, they become the object of their anger. Purcell [38], in agreement with Eldridge and Finnican [34], surmises that the reaction of abusive parents stems from failures and reversals in the early developmental stage of grandiosity because of the parent’s inability to engender in the child an initial feeling of omnipotence and gradual disillusions suitable for bearing the frustrations and limitations imposed by reality.

Steele [39] reports that his extensive clinical experience with abusive parents enabled him to find that they, even when they were not physically abused in their childhood, suffered from a lack of adequate emotional support and, in general, from profound neglect of their emotional needs. As Steele states, “neglect is harder (than abuse) to define, to tell how much love is not there-something like trying to describe the contents of a vacuum” [39] (p. 1006). He points out, in particular, how devastating the absence of maternal validation responses is for the child, an absence that does not allow him/her to develop the basic trust that is essential to sustain and maintain narcissistic balance. The parent who lacks the ability to be empathetic is unable to recognize and validate the child’s emotional experience [40].

Galdston [23] pointed out that it is precisely the parent’s narcissistic fragility that does not allow him or her to deal adequately with the child’s emancipatory behaviors, first and foremost during the early separation–individuation phase, when the child, through assertive behaviors, displays his or her need for individuation. The abusive parent confuses assertiveness with an aggressive attack aimed at his/her fragile self and may have uncontrolled angry reactions. The parent thus communicates to the child that he/she does not tolerate aggression and perceives it as violence. Aggression, here understood as a vital drive aimed at individuation, cannot be tolerated and is punished. At the same time, Galdston [23] observed in his work with 75 abusive families that the abusive parent finds it very difficult to prohibit the child’s improper actions because he/she fears that the prohibition will be followed by the child’s (intolerable) violence. The mother cannot bear the threat of her child’s aggression. In his extensive experience, Galdston found that outbursts of anger and abuse alternate with an inability to put appropriate limits on children’s behavior. He reports, for example, of one mother who constantly moved objects around the living room in a vain attempt to keep the objects out of her 3-year-old daughter’s reach, without ever being able to tell the child directly that she did not want her to touch them. Another mother complained about how difficult it was for her to enter a store with her daughter because she felt obligated to buy the child everything she wanted, because of fear that if she refused, the child would make a scene and embarrass her. Another mother reported being afraid to meet other children for fear that her daughter would see their toys and want them for herself and that she would be unable to say “no.” Among the youngest children, who were just beginning to walk, Galdston observed that the interaction with their mothers was characterized by a description of their aggression completely out of proportion to the facts. The mothers spoke of their children as if they were threatened by their mobility. “Wild”, “uncontrollable”, “destructive”, and “attacking” were among the terms most often used to describe children barely able to take their first steps. Usually, these adjectives were used to modify the term “monster,” which was the epithet most often used to label children. Mothers responded to their children’s first expressions of aggression either by avoiding them through denial or by violent physical abuse. Both of these patterns of response left children with the same impression, namely, that they were carrying dangerous aggression, unmanageable by others and themselves because of its potential for violence.

### 3.2. Therapeutic Intervention

The first question that arises is how to intervene in cases of abuse while ensuring the child’s safety and, at the same time, not compromising the therapeutic alliance with parents. Tuohy [22] observes that sometimes the removal of the child can have an ego-supportive effect if the social worker does not have a punitive attitude, but, on the contrary, this effect can be achieved if he/she shows a genuinely protective attitude by communicating to the parent that he/she cannot allow him/her to harm him/herself and/or the child and that he/she intends to protect his/her family from such danger by taking care of all its members, parents and child. In any case, it is widely suggested to avoid the removal of the child whenever it is possible to provide treatment to the parents [41].

A pioneering intervention is described by Galdston [11] who established The Parents’ Center Project in Massachusetts in 1969 for the dual purpose of protecting children from further abuse and strengthening the integrity of their families. In its first 7 years, the Center treated 46 families with 73 children, being able to care for 15 families at a time. The treatment, which consisted of a therapeutic daycare unit for children and a therapeutic group for parents on a weekly basis for a period ranging from four weeks to five years, proved to be very effective for children, who showed good developmental recovery and was helpful in preventing further abuse, which occurred in only two mild cases for a short period. Galdston noted that the changes in the parents (mostly the mothers were the abusive parents) were not profound; however, caring for the parents and children at the same time allowed for the establishment of a good therapeutic alliance and thus willing acceptance of the care provided to the children. Galdston does not go into the difficulties encountered with the parents; he merely describes their mental functioning dominated by deep ambivalence toward themselves and their children, feelings of profound dissatisfaction in their relationships with their parents and partners, and a pervasive inability to desire and achieve a satisfying life undergirded by intense feelings of guilt and a poor self-image.

Blumberg [42] reports that, in his experience, he has encountered predominantly abusive mothers of often very young children (less than one-year old) and, in any case, usually no older than three years, and he attributes this to the fact that mothers are usually the caregivers of the young child for most of the day and, therefore, the ones most frequently exposed to the child’s demands. Regarding treatment, he suggests not only considering the abusive parent but also the other parent since abuse is a family problem and signals a crisis in the family unit. He also recommends trying not to separate mothers from their children and to house both in residential facilities where trained professionals can take care of both the parent and the child. This intervention, in Blumberg’s opinion, should precede long-term psychotherapy of the parent. Regarding this, he stresses that the therapist should first deal with his or her negative countertransference toward the abusive parent and focus on the parent’s intrapsychic conflicts rather than on the parent–child relationship in order to prevent the narcissistically fragile parent from feeling neglected by the therapist and thus developing hostile feelings toward the treatment. Psychoanalytic psychotherapy would be the best form of therapy for the abusive parent, who usually has a negative self-image and suffers from ego weakness, along with treating the other parent, who, at best, has been unable to offer the necessary support to the abusive parent. When parental hostility is too high and does not allow for a sufficient therapeutic alliance, it can be very helpful to offer parents the opportunity to participate in help–help groups where they can confront other abusive parents without feeling judged and without fear of punishment. Blumberg also finds it very useful that twenty-four-hour “hotlines” run by trained staff are available to accommodate requests from parents in crisis.

Concerning couple dynamics, a very interesting contribution is made by Freedman [41], who stresses how necessary it is to consider abuse a couple issue, while avoiding placing sole responsibility on the abusive parent. She points out that, in her experience, very frequently projective identification considerably characterizes the couple relationship and that this implies that the abusive parent is often the repository of the nonabusive parent’s projections and that it is precisely these projections that are decisive in triggering the abusive behavior. Freedman’s recommendation should be taken seriously, given how widespread the practice of removing the abusive parent from the household without considering the contribution of the nonabusive parent is. Freedman advises against colluding with the couple’s tendency to attribute difficulties to one parent alone, a tendency that is very often supported by the nonabusive parent’s need not to address his or her intrapsychic conflicts in order to avoid intolerable emotional pain. Freedman proposes a multifaceted model of assessment that includes couple interviews, individual interviews, and couple interviews in the presence of the child for the purpose of achieving a psychodynamic understanding of the parental couple that allows for a focus on the deep dynamics that contributed to the abusive behavior and for assessing the possibility of the couple’s involvement in therapeutic work to address the difficulties.

From the study of the articles included in this review, three main types of psychotherapy treatment based on psychoanalysis emerged: parent–child psychotherapy, classical individual psychoanalytic treatment, and mentalization-based treatment.

#### 3.2.1. Parent–Child Psychotherapy

The pioneer of parent–child psychotherapy in family situations of abuse or high risk of abuse was Selma Fraiberg [10,13]. Many of these families at assessment turn out to be multi-problem, hard-to-reach families that are clearly unmotivated to undertake treatment. Selma Fraiberg [10] developed a specific program at the Child Development Project, University of Michigan, where two psychoanalysts, three clinical psychologists, two social workers, and a pediatrician worked, all of whom were part-time, thus representing the full-time equivalent of 3.5 staff members. The team was able to care for more than 140 families per year. In reporting on their first 50 cases, Fraiberg points out that their method combined psychoanalytic principles and techniques with traditional social service practices and theoretical knowledge of developmental psychology.

First, Fraiberg stresses the need to understand the transference dynamics and defensive apparatuses of these parents. She points out that transference, understood as the unconscious repetition of the past in the present in which people from the present are perceived with the qualities of significant figures from the unremembered past, is not only elicited by the psychoanalytic situation but is present in everyday life and manifests itself, especially in emotionally conflicting situations; thus, social workers, who usually meet their patients at times in life when they experience severe emotional conflicts, can be helped to understand their patients if they pay attention to transference dynamics. Fraiberg, for example, describes a young mother who, after being referred to the hospital, did not respond to summonses and did not show up until a month later as a woman who was presumably very frightened about meeting the social worker, imagining her to be a dangerous woman. When the girl showed up, she appeared angry, demanding, oppositional, and kept repeating that no one was helping her and no one understood her. The therapist decided to immediately interpret the negative transference by telling the patient that she well understood her anger toward those who were trying to help her if she did not feel understood; she wished to understand her, but if the patient did not feel understood, she would have reason to be angry with her and could tell her so. In another example, a mother showed up for appointments but would not speak. The therapist assumed that the patient’s silence was a defense against painful affects and, after long silences, she told the patient that she thought she was very angry with her, coming to appointments to be helped without getting any help! No wonder she did not trust her; why should she?

In both cases, the interpretation of negative transference was crucial in fostering the therapeutic alliance. Later, it became clear that both patients were transferring the internal image of their past parents onto the therapist from the very beginning. The risk in such cases is that the patient’s negative transference triggers negative countertransference in the therapist, compromising the treatment outcome. Fraiberg emphasizes how necessary it is for the therapist to ask what motivates the parents’ abusive behavior and to avoid merely using labels. She writes: “Label is not a diagnosis. It is a mailing address” [10] (p. 96), meaning that if the therapist merely labels a mother, for example, as refusing, and the mother does not become nonrefusing, the next address will be the court, and after the court, the mother and child may have different addresses. On the other hand, understanding what leads a mother to reject her child can enable the treatment of the mother and the mother–child relationship by fostering the psychological development of both. In this regard, Fraiberg reports how enlightening it was, in the case of a mother who was evidently rejecting her four-month-old daughter, to understand that the woman had such an unconscious fear of hurting her daughter that rejection was the best way she had found to protect her daughter from herself. In this case, psychotherapy conducted during home visits once–twice a week enabled the mother to be able to acknowledge her own pain experienced in her abusive childhood experiences and to have less fear of being identified with the aggressor and acting against her daughter.

Identification with the abuser is a defense frequently used by abusive parents who were themselves abused in their childhoods: It is the defense that allowed them to defend themselves against distress and ego disintegration through a kind of unconscious psychological fusion with their abuser. If the therapist stands by the patient and allows him or her to experience, within the safety of their therapeutic relationship, the pain that he or she was unable to experience at the time, the risk of acting out abusive behaviors toward the child decreases considerably, while the patient gradually manages to disidentify from the abusive object.

Fraiberg’s working model has continued over time to guide the organization of public services caring for parents or high-risk children. Lieberman and Pawl [7] report a clinical example of the treatment of an abusive mother and her child that shows how effective integration of developmental guidance, infant–parent psychotherapy, emotional support, and practical assistance can be. They emphasize how, in cases referred by the court for treatment, it is necessary to conduct an assessment intervention, including home and office visits, for a few weeks in order to assess the possibility of building the therapeutic alliance with the parent and to avoid imposing psychotherapeutic treatment out of the blue. Lieberman and Pawl also recommend interpreting negative transference from the earliest sessions, and they present a reflection regarding the appropriateness of including the child in sessions with the mother: On the one hand, participating in sessions with the mother exposes the child to highly charged material; on the other hand, if the child has been abused, he or she can be greatly helped by joint sessions with the mother to deal with these situations in a protected and safe context, in which the mother’s disturbing behaviors can find acceptance and can be recognized for their deeper meaning. The therapist’s interventions can help the child to better cope with the situations he or she experienced with the mother.

Selma Fraiberg went on to work at the Infant–Parent Program at San Francisco General Hospital—University of California, which was later directed by Seligman, who integrated infant–parent psychotherapy with nondidactic developmental guidance and concrete support. The Program was able to care for about 60 families per year with children up to three years of age [43]. The intervention usually began with a four-to-six-week assessment of the family in order to learn about the child’s developmental level, the psychodynamics of the parent–child relationship, and the couple relationship, if any, as well as the family’s psychosocial and economic issues. Many cases required cooperation with child protective agencies and courts. Videotaping parent–child interactions for later viewing with the family was also introduced in some cases. Treatment usually took place in a weekly visit lasting 60–90 min at the family’s home. The focus was the infant–mother relationship; however, other family members were included in the treatment program when indicated. According to Seligman, home visiting allows the therapist to learn more about the relationship and makes the parent feel more understood about the demanding task he or she faces even on a practical level in raising the child. At the same time, the therapist has to deal with situations different from those he or she encounters in his or her office, including new transference and countertransference configurations, and sometimes it can be difficult to maintain the therapeutic frame, yet this is often the only way to reach families. Many of the therapist’s efforts are aimed at establishing and maintaining the therapeutic alliance, which sometimes involves continuing to go to the home of patients who do not show up, thus continuing to maintain the trust and hope that the therapeutic relationship can begin and be maintained. Working with these families also involves providing them with direct support by helping them in their contact with other social agencies. The usual analytic interpretive work thus takes place in unconventional contexts. Building a supportive relationship provides the basis for interpretive analytic work that focuses on repetition in the transference to the therapist and to the baby to help parents realize how much they are reenacting their pasts. Within the therapeutic relationship, nondidactic development guidance is conveyed with nondirective techniques designed to help the parent understand the child’s mental states while avoiding the bias of their own negative childhood experiences. Seligman emphasizes how often in these cases the therapist must deal with what he calls bureaucratic transference, meaning the expectations that the parent has developed from previous experiences with the social agencies with which he/she may have previously come into contact. Very often, in fact, abusive parents come to the therapist after having been referred to other social agencies and have sometimes already had contact with the court.

Seligman emphasizes how effective early infant–mother psychotherapy can be precisely because new parents are particularly prompted by their condition to recall their childhood experiences and reflect on their relationship with their parents. Their internal representations and defenses are very evident and much easier for the therapist to understand than during usual psychotherapy with an adult patient. Recent motherhood or fatherhood can be a valuable opportunity for change: While it can awaken old conflicts, it can also be an opportunity for deep processing and change. The therapist’s help can foster good responses from the child that can disconfirm the parent’s expectations and thus enable the child to experience a new beginning, discovering in him/herself unexpected resources and a new and happier possibility of a parent–child relationship, which can free him/her from the unconscious impact of his/her own negative childhood experiences. The therapeutic relationship, with containment and soothing, helps parents counteract the negative internal representations that had begun to deteriorate the relationship with the child and provides the basis for interpretive work. The quality of the therapeutic relationship and interpretive work gradually enables the development of greater self-confidence and disidentification from the bad internal and external objects.

Arons [44] believes that mother–child psychotherapy is organized around goals common to both psychoanalysis and attachment theory, namely, the possibility of recognizing, naming, and metabolizing feelings containing distress and fear. The therapist simultaneously considers the relationship between mother and child; between self and mother; between self and child; and between child, mother, and self. The child communicates through body language and affect with the therapist, whose job it is to be emotionally reached and, subsequently, to transform the affective communications by symbolizing them into words. The therapist tries to hear what the child is expressing, to name the child’s affective communications to keep the child, the mother, and her relationship with both in mind, paying attention to what the child evokes in the mother and what the mother evokes in the child and what both evoke in the child. Arons notes that the therapist is caught between the child’s developmental drives and the mother’s conflict between repetition of the past and hope for a new and different future.

Ludwig-Korner [45] wonders whether infant–parent psychotherapy is a psychoanalytic method. There are many differences between the usual psychoanalytic treatment and mother–child psychotherapy. In the latter, the therapist has a bifocal approach, sometimes including other family members; the setting can change both in relation to the length of the sessions and to the location, taking place sometimes in the therapist’s office and sometimes in the parent’s home. In addition, the goal is to reduce symptoms as quickly as possible; sometimes the therapist and parent may discuss concrete interactions and videotape some interactions in order to discuss them later. The therapist is very involved in the mother–child interaction; he/she tries to capture affective communications and provide them with containment, thus trying to help mother and child develop some specific ego functions, first and foremost, a greater capacity for emotional regulation. Ludwig-Korner questions whether this type of psychotherapy involves the constitutive aspects of psychoanalytic work, such as the analysis of unconscious fantasies, relational representations, and transference. The child has not developed symbolically represented relational fantasies but shows the therapist through his/her emotional reactions what his/her representations of relational experiences are, and sometimes he/she enacts the emotional state of the parent who is talking to the therapist. She concludes that not only is mother–child psychotherapy a psychoanalytic treatment but adds that psychoanalysis has benefited greatly from the empirical contributions of infant research studies and attachment theory, for example, coming to understand that nonverbal emotional exchanges are at least as important in analytic treatment as verbal exchange, that a crucial goal of treatment is to help the patient develop self-regulating ego functions, and that this is accomplished first and foremost through sensitive and responsive listening.

Amanda Jones [46] also points out that early mother–child psychotherapy can offer the mother a valuable opportunity to address deeper conflicts related to dependence, vulnerability, and experiences of helplessness precisely because these are so intensely evoked by the infant. Psychoanalytic foundations can be found in the task of making meaning of observed behavior, hypotheses about the influence of unconscious processes, and interpretive work performed to construct a narrative. The child “remembers” through his/her actions; the parent shows how he/she defends him/herself from emotional contact with intolerable emotions to which the child exposes him/her. Mother–child psychotherapy helps the parent to be able to afford to suffer emotional pain in the company of a reliable therapeutic relationship and helps the child not to identify with the defenses used by the parent. The therapist can observe how both mother and child defend themselves through the use of projective identification and how this can fuel destructive escalation. The therapist’s ability to contain the distress felt by the child and parent helps both develop greater emotional regulation skills and decreases the need for both to resort to the defensive use of projective identification. The child’s distress can find an adequate mirroring response from the parent if the parent succeeds in not being overwhelmed by the child’s distress. If the parent fails in this, on the contrary, he or she will provide the child with an amplified version of his or her own distress by feeding it further and forcing the child to enact maladaptive defenses. The therapist’s ability to contain the distress of both and put into words the emotional happenings experienced allows the parent to become more aware of his or her own distress and defenses [46]. Jones also makes use of videotaping to show the parent the defensive processes in action and to help the mother understand the child’s mental states and translate them into words, thereby increasing her ability to mentalize. She videotapes a few minutes of interaction between the mother and the child to review them with the mother during the next session, inviting her to free associate with what she sees. Observing the videotape can allow one to see that the parent attributes mental states to the child that the child is not experiencing, helping the parent realize that the child makes them his or her own and that these are precisely mental states that the parent does not tolerate in himself or herself. For example, a parent may attribute hostile intentions to a child who wishes to approach playfully, and subsequently, the child becomes angry [46]. In the clinical example presented, Jones shows how a mother realizes, upon viewing the videotape, how frightened she appears to be of her son, realizing that it cannot really be such a small child who frightens her. Later, the mother realizes that she is emotionally confusing the experience with her child with the experience with her abuser in the past. Working in this way, it is possible to make sense of the observable behaviors by connecting them to the unconscious processes that cause them, thus freeing both mother and child from the compulsion to act out destructive behaviors that take the place of what cannot be endured and, thus, thought about. Jones, too, stresses the importance of accommodating negative transference because it is necessary for the therapeutic setting to first allow freedom to express negative feelings so that feelings of love can truly emerge [46].

#### 3.2.2. Individual Psychoanalytic Treatment

The search for literature material has revealed a considerable paucity of contributions regarding the individual psychoanalytic treatment of parents who have abused their children. Only Steele [39] reported his clinical experience with middle-class parents treated in a private setting. Presumably, the paucity of contributions in this area also depends on the fact that child abuse is a public health issue and thus engages public health services in developing specific treatment programs, but public services cannot afford the expense of high-frequency psychoanalytic treatment. However, given instead the considerable production of articles regarding the intensive psychoanalytic treatment of abused patients, it is also plausible to wonder whether the abusive parent is unlikely to require psychoanalytic treatment and/or whether psychoanalysts may be reluctant to offer intensive psychoanalytic treatment to people who have abused their children.

Indeed, some articles point out how difficult it can be for the therapist to come to terms with negative transference and countertransference with these patients.

Tuohy [22] notes that working with abusive parents is narcissistically depleting for the therapist, given that these families have very profound needs, frequent crises, and intense negative transference reactions. Tuohy reports that Altshul [47] pointed out that, in these situations, the therapist may enact some defenses: denying his or her fatigue and, through reaction formation, duplicating efforts or overidentifying with the patient so that he or she may feel that he or she is receiving what he or she is giving the patient. The risk of burnout is very high in the treatment of these patients, and to combat it, Tuohy [22] suggests the constant presence of supervisor support, peer support through frequent case presentations, and the possibility of engaging in other professional activities, e.g., intervention planning, teaching, research, and training. The patient’s anger can strain the therapist’s ability to resist without retaliation; at the same time, to help the abusive parent, it is crucial to make it easier for him/her to tolerate ambivalent feelings in treatment by allowing him/her to have the experience of being able to express his/her anger without the threat of destruction or losing the therapist. Tuohy observes that the therapist runs the risk of denying the abusive part of the parent when he/she fails to address the feelings of anger that the patient communicates.

Green [15] notes that abusive parents are often defiant and masochistic because of their deep, unconscious need to turn the treatment situation into a repetition of their early experiences of rejection. The therapist should be prepared to deal with negative countertransference, which includes the natural tendency to be indignant about the abusive parent’s callous and cruel behavior toward the child. In addition, the abusive parent, who often shows hostility toward the therapist and a lack of cooperation, misses appointments and arrives late, constituting an attack on the narcissistic balance of the therapist, who does not feel acknowledged in his or her efforts.

The analyst must prepare to step into the shoes of a controlling, critical, punishing object that is difficult to trust [39]. The unconscious roles of both victim and abuser are re-created in both intimate relationships and transference, and a paranoid worldview associated with the avoidance of emotional contact and defenses against feelings of helplessness and worthlessness dominates in the early stages of treatment, characterizing transference–countertransference exchanges dominated by feelings of hatred [48].

Some psychoanalysts who have approached the subject of the psychoanalytic treatment of abused patients have highlighted the difficulty of working with their hatred and the perverse structure they have developed by eroticizing hatred.

As Milton [49] states, identification with an abusive object that despises the frightened child’s feelings and sadistically eroticizes pain and hatred is what makes it most difficult for the therapist to treat the patient, who must also deal with the depth of his or her corruption, hatred, and addiction to perverse arousal. At the same time, the perverse structure can provide stability and may be the best psychic compromise solution, fulfilling many functions: It transforms painful experiences into gratifying ones and stabilizes them by creating dependence on the arousal stimulated by cruelty and the accompanying feeling of omnipotence, and, in addition, when the perverse area of functioning is sufficiently circumscribed and split, it can allow tolerable degrees of normal functioning, protecting against psychotic breakdown [49].

Experience with abused patients also teaches that the analyst must be careful to show them both love and hate; the analyst must do this by being careful to show both in small doses: Hate makes the patient feel safe, but it destroys, while love makes the patient feel less persecuted, but only for a while; then, it can trigger persecution. It is a constant give and take. When the emotional temperature is too high, it can sometimes be helpful, even to the establishment of an “as if” dimension in the analytic couple, to observe the similar dynamics (of love and hate) that the patient experiences outside the analytic couple [50].

Winnicott [51] wrote that “whatever his love for his patients, he (the analyst) cannot prevent himself from hating and fearing them, and the more he realizes this the less he will let hate and fear determine what he does to his patients” (p. 235).

Gabbard [52] has dealt extensively with this issue, pointing out how patient hatred can induce in the analyst the temptation to counterattack (through actions or interpretations used as hostile actions), the desire to retreat into detached indifference, or the denial of the hatred itself. He warns that hatred becomes persecutory and destructive in treatment, particularly when it is hidden and denied, so the analyst who tries to offer love as an antidote to hatred only pushes hatred underground and intensifies its persecutory quality. It is necessary for the analyst to ask him/herself what defensive maneuvers he/she puts in place to avoid hating the patient, aware of the patient’s need for an analyst to show him/her that he is capable of feeling hatred and also to tolerate feeling hatred. The patient will only be able to tolerate his/her own hatred if the analyst can also afford to hate him/her [52].

Another defensive tendency is the temptation to collude with the patient’s splitting, focusing only on his good and loving aspects. Ferenczi [53,54] teaches a great deal about the analyst’s difficulties in accommodating negative transference and how the attempt to meet the patient with infinite patience leads the analyst to acknowledge his own hatred of the patient. As Ferenczi already pointed out, it is a thorough personal analysis that can help the analyst to bear the negative transference, which involves first feeling and suffering it on one’s own skin, before turning it into a thought and then into words for the patient.

Gabbard [55], in this regard, observes how analysts, unaware of their hatred, can make boundary violations at different levels, rationalized by considering the patient a person who has experienced severe deficits that require a real relationship with the analyst to improve, and warns us by helping us recognize that what provides a holding environment is more openly dealing with hatred than providing “love,” considering that the abused patient is asking to come to terms with an attachment to an object of hatred and not love [55].

In contrast, Kohutian-trained psychoanalysts, who are much more optimistic about treatment with the abusive parent, emphasize the potential of the development of a self-object transference in the analytic relationship.

According to Green [15], when planning treatment, it is necessary to intervene at three levels: to decrease environmental stressors by providing the mother with support (e.g., through a home visit program and the availability of daycare facilities for the child) so that she can care for the child according to her own possibilities while waiting for these to increase, to help the child to be less distressed, and to offer support to the mother. Green’s idea is that it is important for the parent to have a corrective emotional experience with an accepting, noncritical, and rewarding adult. He also points out that some specific difficulties must be addressed when treating abusive parents: How can a therapeutic alliance be built if the therapist is involved in reporting to the court? How can a parent so fragile in self-esteem accept help without feeling criticized?

Green stresses that it is necessary to gratify the parent before making demands on him or her and that caution should be exercised in proposing psychotherapeutic treatment for the child because the parent, before a therapeutic alliance has been established with the treatment team, may experience the child’s therapist as a rival and thus sabotage the treatment [15].

Given the narcissistic fragility usually suffered by parents who enact abusive behaviors, from a Kohutian perspective, the development of a self-object transference is crucial. Narcissistically vulnerable patients need their distress soothed to avoid turning it to the children. Eldridge and Finnican [34] recommend a noncritical attitude on the part of the therapist, together with an understanding that their abusive behavior indicates an attempt to maintain self-cohesion. This would promote the development of transference. They recommend giving the patient the time he/she needs to expose the split-off, painfully needy self, as well as the time he/she needs to heal the split and not to provide interpretations in the early stages of treatment because they might overwhelm the patient. They suggest strengthening self-object functions while avoiding, however, encouraging regression, providing interpretations of the patient’s wishes and longings so that he/she can experience the parts of the self that have been walled off by the fear of rejection or punishment. According to Eldridge and Finnican, this way of proceeding in therapy by offering him/herself as a self-object allows the patient to feel whole and integrated, which leads him/her to decrease his/her abusive behaviors. Upon examination of the literature considered for this review, only one contribution was found that reports the individual psychoanalytic treatment of abusive mothers within a public service setting in detail [9]. In Brennan’s experience, when children are subject to child protection service providers, rarely do the providers engage with parents to understand the origins of their parenting difficulties to really help them cope with them. According to Brennan, psychoanalytic therapy is the most appropriate treatment because it is the only one that addresses the deeply unconscious origins of the hostility parents act out against their children. However, some issues need to be kept in mind: Not all abusive parents accept the offer of psychoanalytic treatment; there are no empirical studies showing that psychoanalytic psychotherapy is effective in reducing the risk of abusive behavior, and even when it proves effective, it takes a long time for profound changes to occur in the parent; and, finally, it is possible that in the early stages of treatment, the risk of acting out of previously repressed feelings increases [9]. Despite these important issues, Brennan decided to organize a psychoanalytic psychotherapy service for abusive parents moved by a desire to help both parents and children. She observed that the parents who did not accept treatment or stopped it early were the ones who suffered from a deep sense of helplessness and seemed resigned to loss. She believes that these parents found that the best way to deal with their mental pain was to abandon all trust and hope in order to avoid the risk of yet another unbearable disappointment, a defensive maneuver described by Symington [56] as taking refuge in the cocoon.

Brennan presents some clinical exemplifications: with mothers who never showed up for appointments, with mothers who prematurely discontinued treatment, and with mothers who instead pursued it for several years. She highlights how often abusive parents’ stubborn refusal to begin treatment can reveal their unconscious desire not to care for their children in order to protect them from their hostility and in the hope that they will be placed with more competent families. When patients succeed in starting and continuing long-term treatment, the early narcissistic damage suffered has not totally destroyed their hope, probably because of some good relationships they have experienced. Nevertheless, during therapy, Brennan experienced the patients’ terror of becoming emotionally close and perceiving their dependence, also bringing the issue of negative transference and countertransference to the forefront [9]. In the course of psychoanalytic psychotherapy with abusive mothers, it became evident that the children were expected to be potential saviors but were soon experienced as persecutors; those who demonstrate the parent’s wickedness, which, being intolerable, must be projected. Due to the effect of transference, the therapist in the therapeutic relationship takes on the role of the persecutor and, at the same time, feels intensely persecuted by the patient, suffering both the pain of feeling deprived and the pain of feeling deprived. This emotional experience, conveyed through projective identification, allows the therapist to “become the patient” who suffered deprivation in her childhood and who, in spite of herself, inflicted mistreatment on her children [9].

Brennan wonders how much these patients may frighten the therapist because of intense transference pressures and to what extent this leads therapists to avoid deepening transference. I would add, is it perhaps also because of this fear that intensive treatments are not offered but, at best, involve weekly treatments? Brennan still wonders whether many of the failures of psychoanalytic therapies with these patients are due precisely to the therapist’s unconscious resistance to being in deep contact in the transference and countertransference relationship with such high and intense levels of destructiveness. Abusive parents often find themselves facing the impossible task of having to meet the deep needs of their children without being able to do so because no one has ever met their own deep needs, and the analyst, countertransferentially, Brennan observes, finds him/herself trapped in the position of someone who is trying to meet the opposite needs of two generations of dangerously deprived families. It is understandable that, in this emotional situation, the therapist does not feel like embarking on and pursuing a long-term therapeutic relationship that involves degrees of emotional pain that are difficult to bear [9]. The therapeutic relationship is effective if the deep needs of the patient are met, but in such cases, the struggle against destructive instances can be too hard.

Psychotherapeutic work in public service is further complicated by the fact that the patient experiences not only transference with the therapist but also with the institution, and the therapist cannot guarantee absolute confidentiality by having to, at the very least, report whether or not the patient shows up for sessions.

Brennan concludes her paper with a reflection regarding a very complex question: Is parenting therapy for the parents or for the children? How much does the therapist have his patient in mind and how much does he/she have in mind the protection of the patient’s children? How can the patient feel that he/she is in his/her analyst’s mind if he/she feels that the analyst has his/her children in mind first and foremost? How can the analyst manage relationships with other professionals involved in child protection? How can he/she use information about the patient that comes from other professionals and not from the patient him/herself?

Despite all of these important questions and all of these difficulties, Brennan concludes that in her experience the psychoanalytic psychotherapy service offered to abusive parents is unique for people who would otherwise never have the opportunity to be helped to come to terms with their grief, their anger, and their despair, often victims of an abusive past and ruthlessly exposed to the risk of becoming abusive parents themselves [9].

#### 3.2.3. Mentalization-Based Treatment

Findings from the previously cited studies support the development of therapeutic mentalization-based interventions adequately suited to the specific needs of impaired parenting. Mentalization-based treatments, first developed by psychoanalyst Peter Fonagy’s research group at University College London, can be considered extensions of Selma Fraiberg’s pioneering work. Rosso [30] suggested that the prevention policies of welfare services should take into account mentalization deficit, especially if associated with a derogatory state of mind regarding attachment needs, as a major risk factor for failure in parenting. Berthelot and colleagues [32] suggested that focusing on mentalization may be crucial for the well-timed identification of individuals with a history of child maltreatment who are expecting a child, since intervening postpartum with parent–infant dyads may already be a step too late, as findings [57] show that an intergenerational impact of child maltreatment can be observed shortly after birth.

Some studies on mentalization-based treatments have focused specifically on intervention with abusive parents. Schechter and Willheim [58] believe that it is necessary for the therapist to allow the patient to experience what it means to safely consider another person’s mind, showing him/herself that he/she can reflect on the patient’s feelings without him/herself becoming dysregulated. In this way, the therapist provides the patient with external regulation and the support he/she needs to begin to consider the mind of his/her children. By keeping the therapist in mind, the patient gradually develops the ability to tolerate his/her own feelings when confronted with his/her children’s distress. The mutative action does not depend on didactic directions, but on developing the ability to consider his/her children’s minds, without withdrawing out of fear of experiencing negative affects again by feeling helpless and retraumatized. If the therapist has helped the parent develop a greater ability to tolerate and regulate his or her negative affects triggered by the children, the parent will be better able to regulate his or her emotions and thus be more willing to consider the mental states of his or her children [58].

Intervening on emotional regulation is crucial, as affective regulation deficits have been widely found in abusive parents [59], and a recent empirical study [60] confirmed that physically abusive mothers have a specific difficulty in recognizing negative emotions, particularly sadness. The parent’s hypersensitivity and irritability indicate that the development of a normal barrier to stimuli has been compromised, presumably as a consequence of a primary relationship that did not protect against stimuli. In cases where the abusive parent was him/herself abused, not only did his/her mother fail to act as a supplementary stimulus barrier by protecting him/her from stimuli he/she was not yet able to tolerate but she subjected him/her to traumatic overstimulation that impaired the development of emotional regulation capacity [22]. Emotional regulation deficits are also transmitted from one generation to the next: If the mother does not protect her child from stimuli she is unable to tolerate, the child will not develop the normal integrative functions of the ego, will easily cry inconsolably, will suffer from sleep and feeding disorders, and will show general restlessness. He/she will thus be a hard-to-manage child who will further trigger the parent’s anger, and thus, a vicious cycle will be fed [22].

Mentalization-based treatment originates from the observation that abusive parents who themselves were abused children develop ineffective defenses against mental pain related to the abusive experiences suffered in the past and that these defenses do not allow the development of the ability to mentalize and process their emotional pain. On the contrary, they defensively resort to identifying with the aggressor [26,27]. Ensink and colleagues [61] found that it is specifically the ability to mentalize one’s traumatic experiences rather than the general ability to mentalize that impairs parenting.

Several mentalization-based programs for parents have been developed over the past 20 years [62,63], based on both individual and group interventions.

Sadler, Slade, and Mayes [64] developed a manualized home-visiting program for young primiparous mothers called Minding the Baby (MTB), conducted alternately by nurses and social workers once a week from the third trimester of pregnancy until the child’s first year of age and every two weeks until the child’s second year. The nurses were mainly concerned with prenatal care and health education while the social workers were mainly devoted to mental health and psychological issues regarding both the mothers and children; however, both used specific techniques to improve mentalization regarding their own and their child’s mental states.

Baradon and colleagues [65] developed a short-term manualized group treatment called “New Beginnings” aimed at incarcerated mothers with children. Each group, led by an experienced psychodynamic psychotherapist, included up to six mothers and their infants and lasted four consecutive weeks, with two two-hour sessions weekly. In each session, a topic, chosen to activate the attachment system (e.g., pregnancy, one’s own childhood experiences, experiences regarding motherhood), was addressed; time was devoted to playing with the children; and then, the mothers were invited to talk about their children, with particular reference to their mental states.

Suchman and colleagues [66,67,68] developed the Mother and Toddler Program (MTP), aimed at substance-abusing mothers of children up to 3 years of age. The goal of the program, consisting of 12 individual weekly sessions that also included the use of videotaping and guidance about the child’s psychological development, was to help mothers become aware of which situations they experienced with their child that elicited difficulties in them and to more accurately understand their child’s mental states.

None of these studies, however, specifically addressed the treatment of abusive parents. Only two recent articles [69,70] report on individual, couple, and group treatments focused on mentalization with abusive parents. Hoffman and Prout [69] emphasize the importance of an individual or group-focused parenting intervention in helping parents master negative emotions toward their children by taking a reflective, non-judgmental stance toward the parent, focusing on which feelings lead the parent to maltreat their children and which painful emotions trigger the child’s disruptive behavior. Spanking the child is thought of as the result of feelings of helplessness and vulnerability against which the parent defends himself for acting inadequately powerful by physically punishing the child. Thus, spanking is conceptualized as a manifest behavior resulting from the parent’s need to avoid painful emotions, so it could be avoided if the parent is helped to understand which mental states of his or her own and the child’s cause the misbehavior to thus move from an automatic reactive stance to a reflective stance [69].

The therapist’s curiosity about the motivations for the child’s and parent’s behaviors helps the parent be, in turn, curious to understand what mental states underlie these behaviors. This intervention, based on psychoanalytic principles, diverges from cognitive behavioral interventions, in that the therapist does not simply instruct the parent on how to act, but first and foremost strives to understand the parent’s pain and to make sense of it, including through the careful consideration of transference and countertransference dynamics. Embracing and containing the parent’s negative emotions by giving them meaning helps the parent himself develop the tools to master the emotions by which he was previously overwhelmed [69].

A recent study [70] describes a group intervention called Group Attachment-Based Intervention (GABI ©) aimed at very socially isolated and at-risk parents of children up to age 5. Parents may attend three groups per week, but may also choose to attend less frequently. Each group session lasts two hours: In the first hour, each parent is helped by a therapist to observe, tune in to, and reflect on their own and their child’s mental states in the play interaction, while in the second hour, the children remain in a playroom with the therapists, and the parents participate in a parent group led by a therapist. The parent group is focused on reflecting on one’s own mental states, one’s childhood history, and the mental states of the children. Parents and children come together in the last 15 min. Reflective functioning is considered the hallmark of this intervention, which, in some cases, is integrated with intensive individual long-term psychotherapy and other supportive interventions [70].

#### 3.2.4. Evaluation of the Effectiveness of Interventions

In a review regarding interventions with abusive parents aimed at improving the parent–child relationship by promoting attachment security, Valentino [71] distinguishes between short-term (5–16 weeks) and long-term (20 weeks–1 year) interventions with weekly visits and the involvement of the mother–child dyad. Short-term interventions aimed at improving maternal sensitivity were rated positively by parents, but no data are available regarding the actual reinvolvement of these parents in the child welfare system.

With regard to long-term interventions, a Randomized Controlled Trial (RCT) [72] found that child–parent psychotherapy was more effective than both Psychoeducational Parenting Intervention (PPI) and the community standard interventions typically available through the Department of Social Services in fostering child attachment security. In addition, child–parent psychotherapy was more effective in maintaining it one year after the end of the intervention. As highlighted earlier, child–parent psychotherapy is a non-directive intervention aimed at establishing a strong therapeutic alliance with the parent who, in the context of a secure relationship with the therapist, can process past experiences so that they no longer iterate negatively on his or her current relationship with the child. PPI, on the other hand, is an educational intervention aimed at teaching parents how to engage in more positive interactions with their children and specific skills designed to reduce negative behaviors.

In organizing interventions provided by public social and health services, however, costs must inevitably be taken into account. Abusive parents often need multiple interventions, psychological intervention being only one among many. They often live in poverty and need financial support to provide for their basic needs; in addition, they need practical help in managing and caring for their children until their psychological resources grant them a greater ability to care for their children for a longer time. The shrinking affordability of social and health services leads to the development of short-term and less expensive interventions, including in terms of training professionals. At the same time, on a social policy level, it is necessary to consider what can be the devastating consequences of a lack of early intervention focused on the parent–child dyad. Findings from studies in this field [73] indicate that the intervention typically provided to abusive parents based on case management is insufficient to avoid the negative sequelae of maltreatment involving conduct disorders, personality disorders, and social maladjustment, as well as perpetuating the cycle of abuse in subsequent generations.

Within the framework of attachment theory, several short-term intervention programs aimed at abusive parents or those at risk of abuse have recently been developed, which have proven effective, but only for those parents who have not suffered from traumatic childhood experiences and who are not particularly narcissistically vulnerable [74].

Moss and colleagues [73] report that an attachment-based intervention comprising eight weekly home visits was effective in improving sensitivity in a group of abusive parents; however, careful consideration of the study shows that 72% of parents were reported for neglecting and not for physical abuse, and a follow-up was conducted only 10 weeks later. Moran and colleagues [75] reported that short-term attachment-based interventions were not effective in fostering attachment security or maternal sensitivity for adolescent mothers who had disorganized states of mind regarding their childhood attachment experiences and/or who had suffered physical or sexual abuse in their childhoods. Steele and colleagues [76] also observed that having suffered negative childhood experiences moderates the effect of GABI ©, proving itself less effective in these cases. Van der Asdonk and colleagues [74] note that parents who have experienced childhood maltreatment represent a specific group of parents for whom it is more difficult to intervene effectively. The study also found that abusive parents who have undergone more traumatic experiences respond less well to brief intervention, and these findings indicate that it is necessary to identify these parents early in order to provide them with the most appropriate intervention aimed at processing their traumatic experiences as soon as possible.

Finally, it should be noted that this review highlights the absence of studies evaluating the effectiveness of specifically psychoanalytic treatments, and thus, the cited studies are not included among the evidence-based treatment programs for abusive parents.

## 4. Summary and Future Directions

Research has extensively highlighted that most people who have experienced maltreatment in their childhood develop mental disorders, manifest psychosocial adjustment problems, and, in many cases, become maltreating adults themselves [1,2]. Preventing child maltreatment and treating abused children and abusive parents are therefore pressing public health issues.

When a child is at risk of abuse in a family or when abuse has already occurred, a crucial question concerns assessing whether the child should be removed from the family or whether intervention in the family can prevent future abuse. As established by the Children Act in 1989 [8], child development is enhanced by remaining in the family whenever the child’s safety is assured, and social services should strike a balance between protecting children and ensuring that they can remain with their families [9]. Developing prevention and intervention strategies in public health services for the purpose of repairing, whenever possible, the child–parent relationship should be a social priority.

Psychoanalytic studies have focused on understanding the internal world of abusive parents and identifying which interventions can help them become more competent parents. This article attempts to present a narrative review of psychoanalytic studies in this field specifically focusing on the intrapsychic dynamics of physically abusive parents and the treatments developed by psychoanalysts to treat them.

The article focuses specifically on physical abuse, following the recommendation [14] not to group together different types of abuse, since it has long been known that, for example, sexual abuse is an issue completely apart from physical maltreatment [3].

A narrative review was conducted for the purpose of presenting an overview of psychoanalytic studies in this field, focusing on the intrapsychic dynamics of parents who physically abuse their children and the therapeutic intervention.

Regarding intrapsychic dynamics, the review highlighted the predominant role of the transgenerational transmission of abuse and narcissistic fragility. Not all abused persons become abusers themselves, and this depends on how well the experience of abuse was processed and accepted without recourse to denigrating defenses or based on identification with the abuser. Not all abusive parents were physically abused in their childhoods, but even those who did not experience abuse suffered from a lack of recognition and validation from their parents, a lack that resulted in a deep narcissistic vulnerability that leads them, when they, in turn, become parents, to need their children to support their fragile narcissistic balance. They are thus truly psychically ill-equipped to support their children’s needs and they cannot understand children’s aggression as a life drive. Physical abuse, in these cases, follows feelings of helplessness and of vulnerability evoked by their children’s demands that are intolerable to them.

Therapeutic treatments developed in the psychoanalytic clinic thus focus on helping the parent come to terms with past experiences and his or her narcissistic vulnerability. Interventions based on mother–child psychotherapy and mentalization have been found to be prevalent, while there is very little literature regarding intensive individual psychoanalytic treatment.

Research on treatment effectiveness has shown that while brief therapeutic interventions aimed at improving parental sensitivity are effective in many other cases, parents who have experienced traumatic childhood experiences need long-term intensive therapeutic interventions.

Psychoanalytic research has highlighted how psychoanalytic intervention in these cases is very difficult by presenting specific difficulties in managing negative transference and countertransference. No article focusing on the efficacy of long-term individual psychoanalytic treatment was found, and the review also showed that interventions are aimed in almost all cases at mothers, although the need to involve both parents was stressed by several authors.

This review also highlights another important limitation of psychoanalytic studies regarding the treatment of physically abusive parents, namely, the absence of controlled studies on their effectiveness. As mentioned above, the psychoanalytic studies cited herein are not included in the evidence-based treatments. It is believed that future research should first make up for this deficiency. As is well known, the issue of outcome research in psychoanalysis is very complex, yet efforts in this direction are necessary in order for the effectiveness of psychoanalytic interventions to be demonstrated.

Specifically, in the area of treatment for physically abusive parents, research can be further complicated by the fact that public health services often complain that they are not financially equipped to provide long-term treatment and that these parents, especially if they come from low-income households, cannot access treatment in private settings.

This review also pointed out that the identified studies were mostly dated. This leads one to wonder why psychoanalysts have long been much less concerned with understanding physically abusive parents and their treatment, while there are plenty of recent studies regarding abused children and their treatment. From my point of view, it would be desirable for future research in this field to regain vigor and for psychoanalysts to be encouraged to write and publish their studies in this area, considering that, despite the absence of controlled studies, the contribution of psychoanalytic understanding can be useful and can contribute to the treatment of parents by fostering healthier growth in children.

## 5. Conclusions

This review focused on the contribution of psychoanalysis to understanding the intrapsychic dynamics of abusive parents and the development of specific therapeutic interventions. With regard to physical abuse, it was found that only about 5% of the studies deal with assessment and therapeutic intervention with parents, while most studies deal with abused children. Yet, there is ample evidence that early intervention with parents is necessary to avoid the devastating consequences and break the cycle of abuse.

The literature on the father–child relationship is entirely lacking, with almost all work on intervention being focused on the mother–child relationship. There are also very few studies focused on individual psychoanalytic treatment. The near absence of literature on individual psychoanalytic treatment and the paucity of contributions regarding the treatment of abusive parents leads one to wonder whether these people elicit serious negative countertransference reactions in therapists to such an extent that they avoid taking them into treatment. Indeed, the issue of negative transference and countertransference was addressed in most of the reviewed articles.

The review also highlights the absence of controlled studies designed to support the effectiveness of psychoanalytic interventions aimed at physically abusive parents. It is recommended that future research should move in this direction.

## Data Availability

Not applicable.

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
