# Peer review of "Psychoanalytic Interventions with Abusive Parents: An Opportunity for Children’s Mental Health"

_ijerph, 2022, doi:10.3390/ijerph192013015_

Round 1
Reviewer 1 Report
I enjoyed reading your manuscript. It is quite dense, with thoughtful discussions related to the citations chosen. I found myself needing to read through the material a number of times. I have a few suggestions that you might consider related to the manuscript. Thank you for the opportunity to engage in the review process.
S1: It would be helpful for me as a reader to see the research and thought process around the selection of pieces for the narrative review. You might consider adding a PRISMA diagram or some other sort of visual aid that shows the process of elimination and reasoning that led to the number of papers reviewed.
S2: There is a lot to this manuscript and most sections were quite full of information. The section on treatment however seemed less complete and it would help, as a reader, to understand how your selected those specific programs. A search of the California Evidence Based Clearinghouse for programs that target parent training for parents who are at risk of, or who have, abused their children show 11 programs, none of which are part of this review. The programs you do discuss Minding the Baby and New Beginnings are not on the clearinghouse and I find little online about these being evidence-based.
S3: I do not recall the Cicchetti et al piece discussing the PCIT program at all - the randomized control trial uses a community standard and a normative comparison - no PCIT. If you did not mean PCIT - Parent Child Interaction Therapy (an evidence based program to increase parent child attachment) then please adjust the language accordingly.
S4: I believe that this manuscript can be clarified that the interventions discussed were the only ones that came through the initial review and discussed as such. I do think the section on interventions would otherwise need to be expanded to discuss some of the extant evidence based interventions that are being used.
Reviewer 2 Report
This article reviewed the psychoanalytic studies in this field, specifically focusing on the intrapsychic dynamics of physically abusive parents and the treatments developed by psychoanalysts to treat them. The following comments of the manuscript can be considered for its revision.
1. More clarification is needed on why physically abusive parents and their interventions should be studied from a psychoanalytic perspective.
2. The article provided an overview of existing therapeutic intervention, but did not identify the gaps of the existing studies and the directions of the future research.
Round 2
Reviewer 1 Report
I feel that the author(s) were able to adequately respond to my suggestions. Thank you.